# Ballast Flow Characteristics of Discharging Pipeline in Shield Slurry System

**Yang Wang [1,2], Yimin Xia [1,2,\*], Xuemeng Xiao [1,2], Huiwang Xu [3], Peng Chen [3] and Guiying Zeng [1,2]**

1 School of Mechanical and Electrical Engineering, Central South University, Changsha 410083, China; 163701020@csu.edu.cn (Y.W.); 183701026@csu.edu.cn (X.X.); 173714177@csu.edu.cn (G.Z.)
2 State Key Laboratory of High-Performance Complex Manufacturing, Central South University, Changsha 410083, China
3 China Railway 14th Construction Bureau Co. Ltd., Jinan 250014, China; hwxcrcc@outlook.com (H.X.); pccrcc@outlook.com (P.C.)
\* Correspondence: xiaymj@csu.edu.cn; Tel.: +86-0731-88876926

**Abstract:** We adopted two-way coupling of discrete and finite elements to examine the non-spherical ballast flow characteristics in a slurry pipe system during a shield project. In the study, we considered the slurry rheological property and the flake shape of the ballast. A ballast size between 17 and 32 mm under different slurry flow rates and ballast volumetric concentration conditions was investigated for determining the law through which the mass flow rate, detained mass percentage, and ballast distribution state are influenced. The results indicate that increasing slurry flow rate and the ballast volumetric concentration increase the mass flow rate; the influence of the latter is stronger. Increases in both in the slurry flow rate and the ballast volumetric concentration can reduce the detained mass percentage in the slurry discharging pipeline, whereas increasing the ballast size has the opposite effect. The increase in both the slurry flow rate and the ballast size changes the ballast motion state. Experiments verified the numerical lifting model of the ballast in the vertical pipeline. The measurements of the actual pipeline wall thickness verified that the simulation results regarding the ballast distribution were accurate.

**Keywords:** slurry shield; slurry system; ballast; mass flow rate; detained mass percentage; distribution state

## 1. Introduction

As large-scale underground tunneling equipment, the slurry shield machine has been widely used in cross-river tunnels, water projects, and urban rail transit engineering [1–3]. The shield slurry system is key to ensuring the stability of the excavating face and the degree of ballast transportation, and the system is mainly composed of a feeding line section and a discharging line section, as shown in Figure 1. Each pipeline section is equipped with a flow meter, density meter, pressure gauge, and pump. The fresh slurry passes through the feeding line section to the cutter head excavation area through pump P1.1, and pumps P2.1, P2.2, and P2.3 absorb the ballast mixture from the cutter head face area and then carry it to the slurry recycling station via the discharging line section. The ability of the slurry system to carry the ballast mixture impacts the shield excavation and its working efficiency. Inadequate ballast carrying capacity can lead to serious engineering accidents, such as pipeline stagnation, blockage, and excavation chamber blockage. The discharging pipeline section is composed of horizontal, inclined, and vertical pipelines. The total length of the pipeline can reach several kilometers and the ballast flow characteristics differ in different sections. The ballast size of an individual particle differs on the millimeter-scale to 400–600 mm, which occurs in the Lanzhou

gravel stratum in China. As such, exploring the ballast carrying performance and the flow characteristics in the slurry system under complex geological conditions is required.

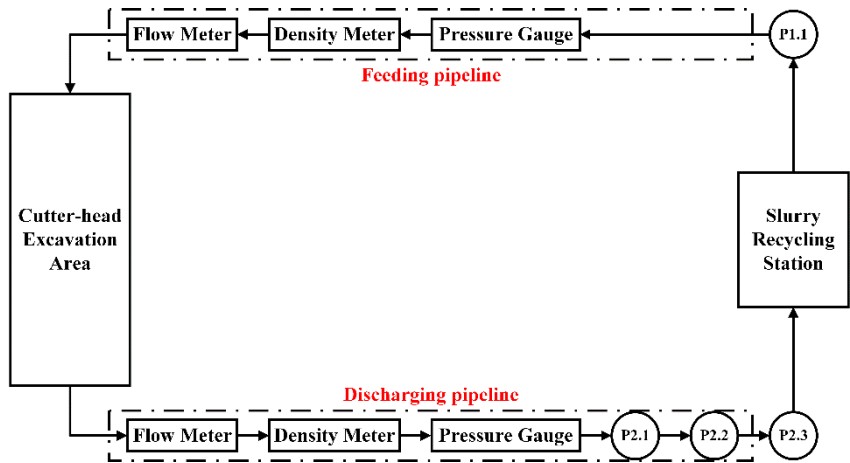

**Figure 1.** Working principle diagram of a shield slurry system. P, pump.

Many scholars conducted experiments and simulation studies on the particle motion in the pipeline system. Li-an et al. [4], Van Wijk et al. [5], Zouaoui et al. [6], and Ravelet et al. [7] established a pipeline circulation experiment system, with a tube diameter of 40–150 mm and extension distance of 5–25 m, and discussed the critical slip velocity of 5–85 mm spherical particle in the pipeline, the particle motion state, and the pressure loss characteristics. So far, no experimental platform has been formed to the shield slurry system, and the experimental results of the published studies have varied. The rules that were obtained by the experiments were often limited and needed to be modified. The particle moving velocity, distribution state, pressure scatter, flow rate, and other information in the pipeline system have been determined while using high-speed cameras, flow meters, pressure gauges, and other methods [8–10]. During the experimental process, purchasing the monitoring equipment is often necessary for obtaining particle movement information. The experiment is expensive, the monitored pipe has a simple structure, and the conveying medium in the pipeline is water, which is quite different from the actual flowing medium. Others established the numerical transportation model for particle sizes between 0.1 mm and 20 mm in the pipeline system based on fluid dynamics, coupled with the discrete element methods, and quantitatively extracted the data, such as particle velocity, distribution region, solid phase volume fraction, and pressure loss of the pipeline [11–15]. The melting phenomenon of ice slurry in the pipe, the mixing effect of solid particles in the viscous liquid, the mechanism of jigging separation of mineral particles, and the cleaning process of non-spherical solid particles in the well hole with the help of a numerical simulation have been investigated [16–19]. By comparison, the numerical method has unique advantages in some respects, such as model accuracy, applicability, completeness, and cost, hence it is suitable for researching particle transportation in a complex pipeline structure.

In this study, we constructed a non-Newtonian fluid model of slurry and non-spherical ballast motion based on numerical simulation; we analyzed the influence of slurry flow rate, ballast volumetric concentration, and ballast size on the ballast mass carrying capacity and distribution state in the discharging pipeline section. The accuracy of the numerical model was verified while using a vertical pipeline hydraulic lifting experiment, and the pipeline wall thickness was measured on-site, which verifies the law of ballast distribution in the simulation results. Therefore, a new and feasible method is proposed here for studying the ballast flow characteristics in a slurry system under complex geological conditions.

The remaining parts of this paper are organized, as follows: The mathematical models of the slurry, ballast, and the numerical model of the slurry system are respectively introduced in Section 2. The numerical method is verified by the experimental data that were obtained from the hydraulic lifting platform in Section 3. In Section 4, the influence law of ballast carrying performance is

analyzed. The ballast distribution state in the simulation is verified through the engineering data in Section 5. Section 6 summarizes the research results.

## 2. Simulation Model of Ballast Motion in Slurry Discharging Pipeline

In research, slurry is treated as a continuous and incompressible liquid medium, and the ballast is assumed to be a discrete solid medium. The crushing of ballast is not considered in the existing model, since the ballast medium is generally rock, pebble, and other solid particles that are not easily broken. The study tends to explore the flow characteristics and the location of the ballast, so we did not consider energy interaction process in the mixture. The rheological parameters of the slurry were tested while using a six-speed rotating viscometer, and a non-Newtonian rheological model of the slurry was constructed. We established the stress model of non-spherical ballast based on the theory of solid–liquid two-phase flow.

### 2.1. Slurry Flow Model

The conservation of mass and the conservation of momentum laws are followed during the slurry flow process. The time-mean Navier–Stokes equation is used to solve the slurry velocity and pressure distribution in the slurry system. The expression of the conservation of mass equation is expressed, as follows:

$$\frac{\partial}{\partial t}(\alpha_f \rho_f) + \nabla(\alpha_f \rho_f \mathbf{u}_f) = 0 \tag{1}$$

$$\frac{\partial}{\partial t}(\alpha_f \rho_f \mathbf{u}_f) + \nabla(\alpha_f \rho_f \mathbf{u}_f \mathbf{u}_f) = -\alpha_f \nabla p + \alpha_f \nabla \boldsymbol{\tau}_f + \alpha_f \rho_f \boldsymbol{g} + \mathbf{F}_j^f \tag{2}$$

where $\alpha_f$ is the slurry volume fraction, $\rho_f$ is the slurry density, $\mathbf{u}_f$ is the slurry flow velocity, $\boldsymbol{\tau}_f$ is slurry shear stress, $p$ is slurry pressure, $\boldsymbol{g}$ is the acceleration of gravity, and $\mathbf{F}_j^f$ is the is force between the $j$ th ballast and slurry in a single grid volume.

The slurry that is used in the system is composed of bentonite, pulping agent, and water, which is different from the water in terms of density and viscosity. The rheological property of the slurry will have a stronger impact on the force of the ballast and the velocity distribution of slurry. The slurry density that was extracted from the site was 1100 kg/m³, and Figure 2 shows the slurry rheological characteristic curve.

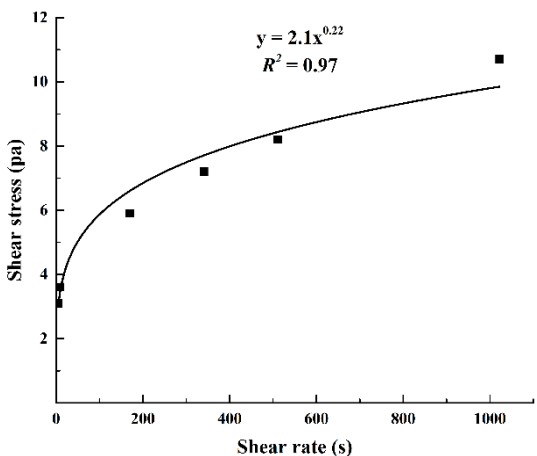

**Figure 2.** The rheological curve of slurry.

The slurry is pseudoplastic fluid based on the rheological curve. The shear rate and shear stress meet the power law rheological property model, shown as follows:

$$\tau_f = k\dot{\gamma}^n \tag{3}$$

where $\tau_f$ is shear stress, $\dot{\gamma}$ is shear rate, $k$ is slurry viscosity coefficient, and $n$ is the power law index, the values of which are listed in Table 1.

**Table 1.** Slurry rheological parameters.

| Parameter | Value |
|:---:|:---:|
| $k$ | 2.10 |
| $n$ | 0.22 |

### 2.2. Ballast Motion Model

In the model, the ballast shape is a non-spherical particle in which the large type is flat chip, and the small type tends to be spherical. A random sample was used for the ballast within the tunneling distance from 120 to 1800 m (Figure 3a). Figure 3b shows the grading diagram of the ballast size that was obtained by the screening method. The figure shows that about 80% of the ballast is within 10–40 mm and the density of the ballast is between 2500 and 2750 kg/m$^3$.

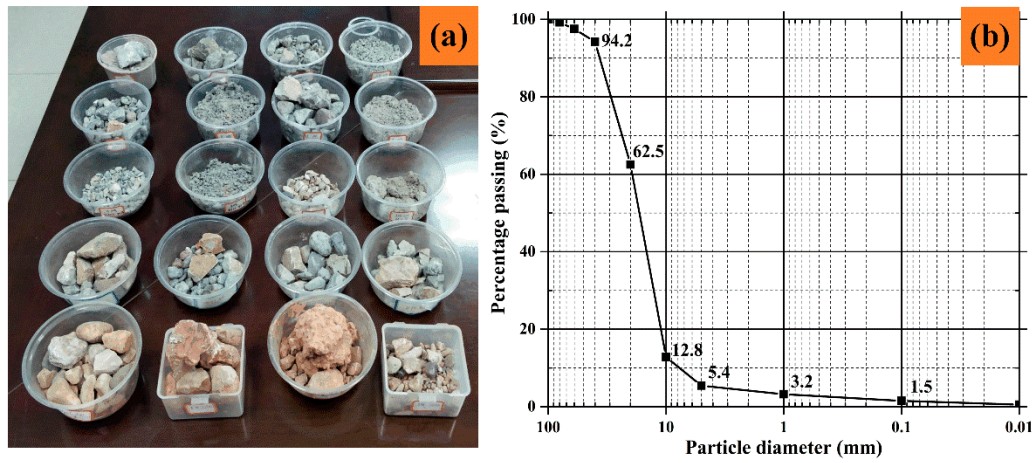

**Figure 3.** Ballast size distribution: (**a**) the ballast sample collected on site and (**b**) the ballast size grading diagram.

We used the Wadell [20] shape coefficient to define the shape of ballast, which is defined, as follows:

$$\varphi_b = A_S/A_b \tag{4}$$

where $\varphi_b$ is the ballast shape coefficient, $A_S$ is the surface area of a sphere of the same volume of the ballast, and $A_b$ is the surface area of the ballast.

Based on the field investigation, the shape coefficient of the ballast transported in the slurry system was mainly in the range of 0.7 to 0.9. Table 2 provides the three-dimensional size, volume, and surface area of ballast.

**Table 2.** Ballast shape parameters.

| Sample | L(mm) | W(mm) | H(mm) | $d_e$ (mm) | $V_b$ (m³) | $A_b$ (m²) | $\varphi_b$ |
|:---:|:---:|:---:|:---:|:---:|:---:|:---:|:---:|
| a | 19 | 14 | 11.5 | 17 | $2.51 \times 10^{-6}$ | $4.58 \times 10^{-4}$ | 0.8 |
| b | 30 | 24.5 | 11.6 | 22 | $5.57 \times 10^{-6}$ | $1.12 \times 10^{-3}$ | 0.78 |
| c | 33.5 | 25 | 13 | 27 | $1.04 \times 10^{-5}$ | $3.0 \times 10^{-3}$ | 0.77 |
| d | 37 | 37 | 15 | 32 | $1.68 \times 10^{-5}$ | $4.29 \times 10^{3}$ | 0.75 |

We divided the non-spherical ballast into four types on the basis of particle size, which is formed by the accumulation of 4 mm spherical particles. Figure 4 depicts the particle size of the ballast and the number of accumulated spherical particles.

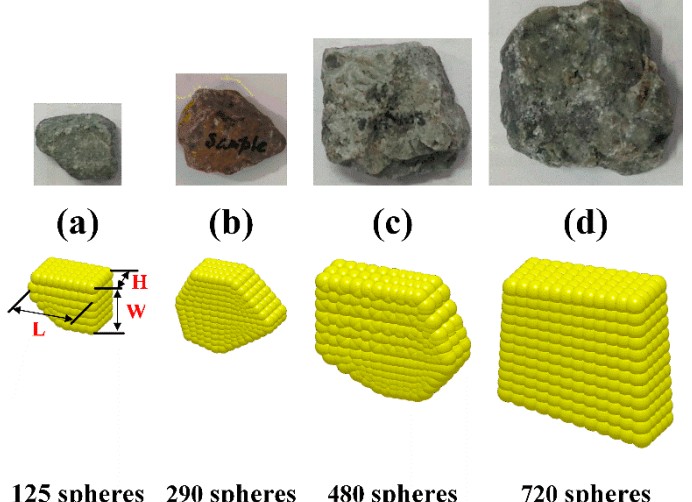

**Figure 4.** The numerical model of ballast sample when $d_e$ = (**a**) 17, (**b**) 22, (**c**) 27, and (**d**) 32 mm.

During the numerical solution process, the ballast is regarded as a discrete phase and the ballast stress follows Newton's second law. The ballast movement in the slurry system consists of the translation motion and rotation motion. The stress model of the $j$ th ballast in the slurry is [21]:

$$m_j \frac{d}{dt}(\mathbf{u}_j) = m_j(1 - \rho_f/\rho_j)\boldsymbol{g} + \sum_{i=1}^{n}\left(\mathbf{F}_{c,i}^{j} + \mathbf{F}_{d,i}^{j}\right) + \left(\mathbf{F}_{c,j}^{w} + \mathbf{F}_{d,j}^{w}\right) + \mathbf{F}_{j}^{f} \tag{5}$$

The moment balance model of the $j$ th ballast during rotation is:

$$\frac{d}{dt}\left(\mathbf{I}_j\omega_j\right) = \sum_{i=1}^{n}(\mathbf{T}_i^{j}) + \mathbf{T}_w^{j} + \mathbf{T}_{DT}^{j} \tag{6}$$

where $m_j$ represents the mass of the $j$ th ballast; $\mathbf{u}_j$ and $\omega_j$ represent the translational and rotational velocity of the $j$ th ballast, respectively; $\mathbf{I}_j$ represents the inertial mass of the $j$ th ballast; $\mathbf{F}_{c,i}^{j}, \mathbf{F}_{d,i}^{j}$, and $\mathbf{T}_i^{j}$ represent the contact force, viscoelastic force, and torque generated by $j$ th ballast and the surrounding ballast $i$ ($i$ = 1, 2..., $n$), respectively; $\mathbf{F}_{c,j}^{w}$, $\mathbf{F}_{d,j}^{w}$, and $\mathbf{T}_w^{j}$ represent the contact force, viscoelastic force, and torque generated by the interaction between the $j$ th ballast and the inner wall of the pipe, respectively; and, $\mathbf{T}_{DT}^{j}$ represents the moment of motion of the $j$ th ballast.

The contact force, viscoelastic force, and stress moment between the ballast and pipeline wall and ballast follow the Hertz–Mindlin no-slip model [22]. The interaction force $\mathbf{F}_j^{f}$ between the ballast and slurry contains drag force $\mathbf{F}_D$, pressure gradient force $\mathbf{F}_P$, virtual mass force $\mathbf{F}_{vm}$, and Magnus rotating lifting force $\mathbf{F}_M$. Table 3 provides the force expressions.

**Table 3.** The interaction force between the ballast and slurry.

| Force | Symbol | Correlation | References |
|---|---|---|---|
| Drag | $\mathbf{F}_D$ | $C_j\left(\mathbf{u}_f - \mathbf{u}_j\right)$ | Haider and Levenspiel [23] |
| Pressure Gradient | $\mathbf{F}_P$ | $-V_b\nabla p$ | Jackson [24] |
| Virtual Mass | $\mathbf{F}_{vm}$ | $0.5C_{vm}\rho_f V_b(\dot{\mathbf{u}}_f - \dot{\mathbf{u}}_j)$ | Oda and Iwashita [25] |
| Rotation Lift (Magnus) | $\mathbf{F}_M$ | $0.5A_p C_{RL}\rho_f \dfrac{\|\mathbf{u}_f - \mathbf{u}_j\|}{\|\Omega\|}\left[(\mathbf{u}_f - \mathbf{u}_j) \times \Omega\right]$ | Schwarzkopf et al. [26] |

Note: $C_j$, slurry–ballast exchange coefficient; $C_{vm}$, virtual mass factor; $C_{RL}$, coefficient of rotational lift; $\Omega$, relative ballast–slurry angular velocity; $A_p$, projected ballast surface area.

### 2.3. Numerical Model of Slurry System Pipe

The pipeline section being studied starts at the outlet of the P2.3 pump, ends at the underlying surface, and consists of three parts: inclined pipeline, horizontal or nearly horizontal pipeline, and vertical pipeline. The P2.3 pump is located at the 259 segment rings away from the tunnel entrance, and the width of each segment ring is 1.2 m. The length of the inclined pipeline is 2.5 m, the vertical pipeline is 21 m, and the horizontal pipeline is over 300 m, which account for over 90% of the total length of the slurry pipeline system. Figure 5 marks the corresponding structures. In line with the calculation capacity of the workstation and research requirements, when considering the similarity of the ballast flow characteristics in the extension part of the pipeline, the size of horizontal pipeline was reduced, while the sizes of the vertical and the inclined pipelines were retained, and Table 4 shows the structural parameters of each pipeline section.

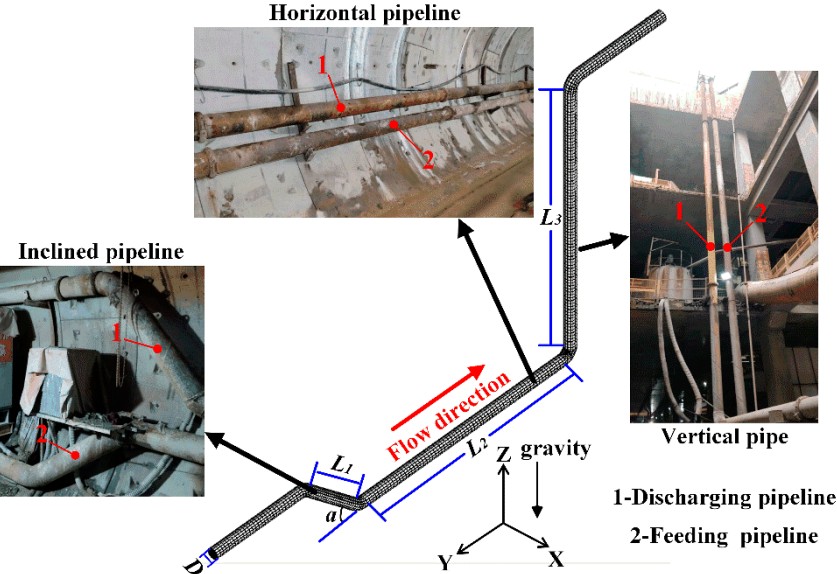

**Figure 5.** The numerical model of pipeline in slurry system.

**Table 4.** Slurry system pipeline structure parameters.

| Parameter | Value | Unit |
|-----------|-------|------|
| $D_p$ | 0.3 | m |
| $L_1$ | 2.5 | m |
| $L_2$ | 30 | m |
| $L_3$ | 21 | m |
| $\alpha$ | 45 | ° |

The numerical model of the slurry pipeline system is divided into three parts: inlet, outlet, and wall. Figure 5 indicates the slurry flow direction and gravity direction. The inlet surface of the pipeline is set as the velocity inlet. We assumed that the velocity of this surface is evenly distributed along the section of the pipe, and the inlet velocity was determined in line with the actual condition. The outlet surface was set as the free stream. The non-slip wall model was adopted. The numerical model adopts the hexahedral structured mesh division. For calculation accuracy, the minimum mesh size must be larger than the particle size. The smallest mesh division size was 10 mm and the number of mesh divisions was 66,056.

Figure 6 shows the flow chart of the CFD–DEM coupling procedure. Firstly, the pressure and velocity in the fluid were solved using the CFD model. Subsequently, the CFD results were transferred to the two-way coupling method (CFD–DEM) to calculate the forces on the particles, such as the drag force $F_D$, rotation lift force $F_M$, and pressure gradient force $F_P$. The position and velocity of particles were solved and updated based on Newton's second law. Finally, the particle

volume information and forces on the fluid from particles were transferred to the CFD solver. This is a complete calculation process and subsequent calculations are performed in this cycle.

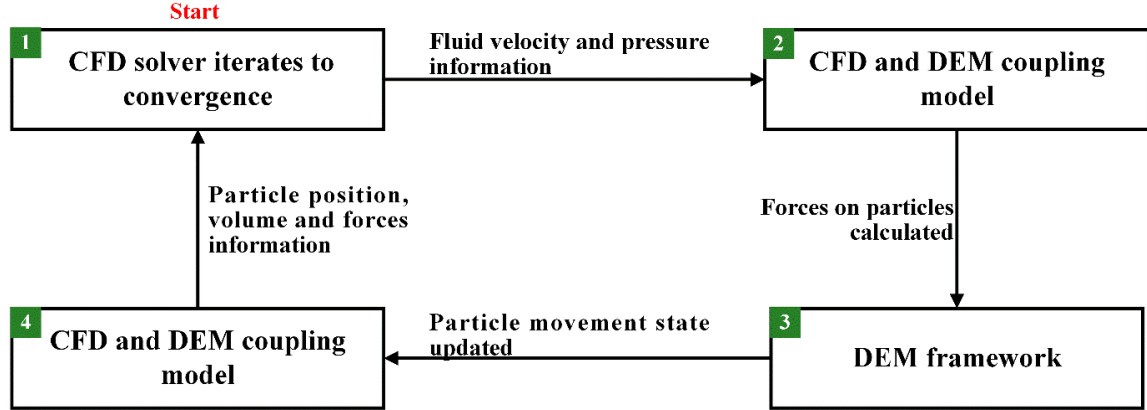

**Figure 6.** Computational flow chart of two-way coupling method (CFD–DEM) coupling procedure.

The slurry flow model was solved while using the semi-implicit method for pressure linked equations (SIMPLEC) algorithm, and the ballast motion model was solved using the explicit time integral method (central difference method). The time steps were set to 0.005 s and 0.0001 s in CFD (Fluent12.0, ANSYS Inc., USA and 2013) and DEM software (EDEM2.2, DEM Solutions Ltd., Britain and 2014), respectively, to ensure the model convergence. The simulation time of each working condition was 50 s, with a total of 64 groups. Each group required nearly 14 h on average for the data processing. The calculation was conducted by a 32-core Intel Gold 6103 (2.2 GHz) and 128 GB RAM workstation computer, which required approximately 45 days in total.

## 3. Numerical Model Validation

The ballast flow characteristics and the suspension velocity were computed and compared with the experimental data, which were acquired from the hydraulic lifting experiment platform, for verification of the ballast motion model. Table 5 list the parameters that were used for the numerical model.

**Table 5.** Data used for verification of numerical model and experiment.

| Parameter | Value | Unit |
|:---:|:---:|:---:|
| $L_3$ | 4 | m |
| $D_p$ | 200 | mm |
| $d_b$ | 10, 20, 30, 40 | mm |
| $\rho_b$ | 1860 | kg/m$^3$ |
| $\rho_f$ | 980 | kg/m$^3$ |

In the experiment, the ballast was pre-stacked on the top of the screen in the vertical pipeline, and by adjusting the working frequency of the pump, we observed a change in the slurry flow rate in the vertical pipeline, and the ballast motion state in the vertical pipeline was recorded by a high-speed camera (Phantom V310, Vision Research Inc., New Jersey, USA) during the flow rate from low to high. The test platform was composed of a pump, a flowmeter, a plexiglass tube, a blanking box, and a feeding machine. The length of the vertical lifting pipe section was 4 m, the plexiglass pipe in the observation section was 2 m, and the diameter of the pipe section was 200 mm. Figure 7 shows the specific structure.

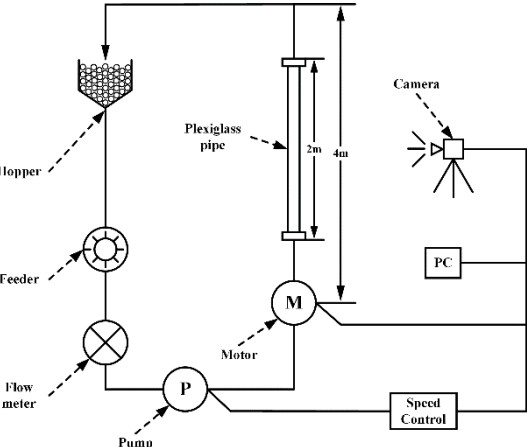

**Figure 7.** Schematic diagram of hydraulic lifting test bench for vertical pipe.

The ballast group adopted spherical particles of uniform size with ballast density of 1860 $kg/m^3$, weight of the ballast group of 10 kg, and the conveying medium in the pipeline was water to facilitate observation and data acquisition. A high-speed camera was used to capture the motion state of the ballast group in the plexiglass pipe section under different flow conditions, as shown in Figure 8. We concluded that the motion state of the ballast group in the simulation model under different flow conditions is basically consistent with the observed results in the experiment. Taking a 20 mm ballast as an example, when the water flow rate in the vertical pipeline is 26 $m^3/h$, the ballast accumulated above the screen is in a static state. When the flow rate increased to 51 $m^3/h$, the ballast group changes from a static state to a slowly rising state, with the upper ballast in a free state and the lower ballast in a concentrated distribution. The ballast was distributed in the pipeline and entered the suspension state when the flow rate reached 74 $m^3/h$. The suspension velocity was 0.65 m/s in the experiment. In the hydraulic lifting numerical model, when the flow rate was 80 $m^3/h$, the ballast group was evenly distributed in the whole pipeline. At this time, the corresponding ballast velocity was 0.7 m/s.

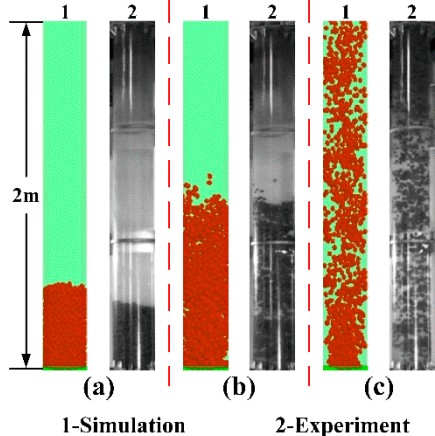

**Figure 8.** Comparison diagram between simulation and experiment on the motion state of ballast with size of 20 mm when $Q$ = (**a**) 26, (**b**) 51, and (**c**) 74 m³/h.

Similarly, the suspension velocities of ballast group with sizes of 10, 30, and 40 mm were obtained, as shown in Figure 9. The maximum difference between the experimental and simulation data was 8%. The simulation results are consistent with the experimental results.

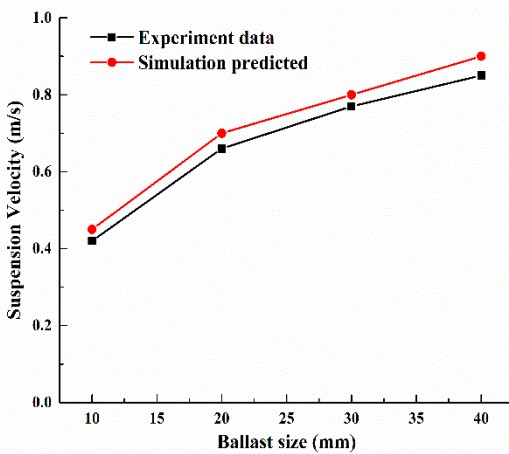

**Figure 9.** Comparison of simulation and experimental data of ballast group suspension velocity under different particle sizes.

## 4. Ballast-Carrying Performance and Distribution State in Slurry System

The project requires ballast carrying capacity and transportation stability of the slurry system. Therefore, we studied the influence of slurry flow rate, ballast size, and volumetric concentration on the ballast carrying performance and the distribution state in the slurry pipeline system. In accordance with the field conditions, the slurry flow rate was 636–1400 $\text{m}^3/\text{h}$, the ballast size was 17–32 mm, and the ballast volumetric concentration was 1.5–6%. Table 6 shows the values of each parameter.

**Table 6.** Value of variables.

| Variable | Values | Unit |
|---|---|---|
| $Q$ | 636, 891, 1145, 1400 | m³/h |
| $d_b$ | 17, 22, 27, 32 | mm |
| $C_v$ | 1.5, 3, 4.5, 6 | % |

During the operation of the slurry system, the amount of the ballast transported per unit time and the detained mass of the ballast in the slurry system must be considered, because the amount of ballast transported per unit time determines the shield machine tunneling speed, and the ballast detained mass negatively impacts the conveying stability of the slurry system. We used the following concepts to quantitatively study the ballast carrying capacity and flow characteristics in the slurry pipeline system, according to the relevant research [6,7,11]. We set up multiple monitoring surfaces in the pipeline sections to acquire the data, as shown in Figure 10.

The mass flow rate $K_m$ is defined as the mass growth rate of pipeline section I, H, and V in unit time after the slurry system enters stable transportation:

$$K_m = \Delta m_{\text{total}}/\Delta t \tag{7}$$

The detained mass percentage $R_m$ is defined as:

$$R_m = m_{\text{residue}}/m_{\text{total}} \times 100\% \tag{8}$$

where $m_{\text{residue}}$ refers to the quality of ballast remaining in pipeline section I, H, and V during the simulation period; and, $m_{total}$ refers to the total quality of ballast that flows through each pipeline section.

The distribution state of the ballast is determined as the ballast distribution area and diffusion degree at sections $I_2$, $H_2$, $V_1$, and $V_2$ of the slurry system. The pipeline was divided into 16 equal parts along the cross-section, in which the inclined and horizontal pipelines were numbered 1–16 clockwise from the bottom of the pipeline, and the vertical pipeline was numbered clockwise from

the middle of the pipeline near the slurry impact side, to facilitate quantitative analysis and compare the results. The diffusion degree is a ratio that considered the area occupied by the ballast and circular cross-sectional area.

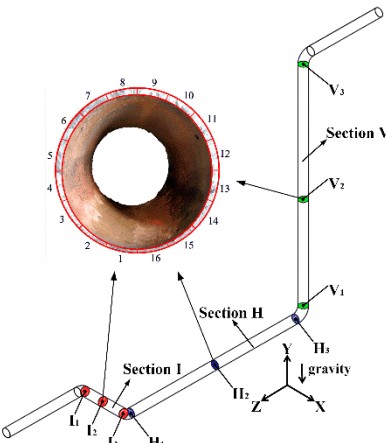

**Figure 10.** Schematic diagram of monitoring surface for each pipe section.

### 4.1. Influence Law of Ballast Mass Flow Rate

The ballast mass flow rate $K_m$ in the inclined, the horizontal, and the vertical sections is the same based on the statistical data when the ballast transportation in the slurry pipeline system enters a steady state. The main factors influencing $K_m$ are slurry flow rate and ballast volumetric concentration, whereas the ballast size has little influence. When the ballast size is 32 mm, the rule of the influence of slurry flow rate on $K_m$ is as shown in Figure 11a. When the slurry flow rate increased from 636 to 1400 m³/h, we observed a linear growth trend under different ballast volumetric concentrations, increasing by 40%, 79%, and 120%, respectively. When $C_v$ was 6%, $K_m$ increased from 31.1 to 68.4 kg/s, showing the greatest increase. Figure 11b shows the influence rule of the ballast volumetric concentration on $K_m$ under the same working conditions. Similar to the slurry flow rate influence law, increasing ballast volumetric concentration also shows a linear growth trend with $K_m$. When ballast volumetric concentration increased from 1.5% to 6%, $K_m$ increased by 104%, 210%, and 322% under different slurry flow rate conditions. Among them, when the $Q$ was 1400 m³/h, $K_m$ increased from 16.2 to 68.4 kg/s, reflecting the maximum increase. By comparing the slurry flow rate and the ballast volumetric concentration, we found that both can improve the mass flow rate.

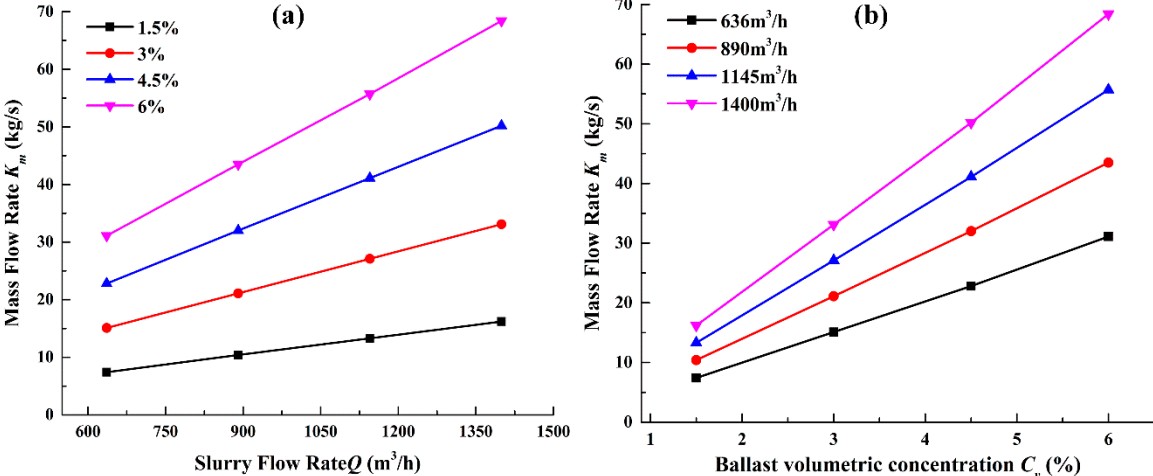

**Figure 11.** Factors influencing ballast mass flow rate: (**a**) the effect of the slurry flow rate and (**b**) the effect of ballast volumetric concentration.

### 4.2. Influence Law of Ballast Detained Mass Percentage

The statistics were calculated on the total mass $m_{total}$ and the residual mass $m_{residue}$ of ballast conveying in the pipeline during the 50 s simulation period. We found that the ballast volumetric concentration, slurry flow rate, and ballast size significantly affect the ballast detained mass percentage $R_m$ of the horizontal pipeline, somewhat influence the vertical pipeline, and basically do not influence the inclined pipeline. Therefore, we only studied $R_m$ in the horizontal pipeline. Figure 12 shows that increasing the ballast volumetric concentration and the slurry flow rate can reduce $R_m$, whereas the ballast size has the opposite effect. When the $Q$ was 636 m³/h and the $C_v$ increased from 1.5% to 6.0%, the $R_m$ varying curve of the horizontal pipeline under different ballast sizes was as shown in Figure 12a. When the $d_b$ was 32 mm, $R_m$ decreased from 55.8% to 29.4%, which is the most significant drop. An increase in the $C_v$ increased $m_{total}$ and $m_{residue}$, and it also increased the probability of interaction with the ballast to improve the carrying capacity of the pipeline. When the $d_b$ was 32 mm and the $Q$ increased from 636 to 1400 m³/h, the $R_m$ varying curve of horizontal pipeline under different ballast volumetric concentrations was as is shown in Figure 12b. When the $C_v$ was 1.5%, $R_m$ decreased from 55.8% to 7.0%, which was the biggest decrease. Increasing the slurry flow rate can increase $m_{total}$ and reduce $m_{residue}$ in each pipeline section, thereby significantly reducing the $R_m$ of each pipeline. Increasing the slurry flow rate can weaken the negative impact of $C_v$ and $d_b$ on the ballast detained mass percentage. When the $Q$ was 636 m³/h and the $d_b$ increased from 17 to 32 mm, the $R_m$ varying curve of the horizontal pipeline under different ballast volumetric concentration conditions was as shown in Figure 12c. When the $C_v$ was 1.5%, the $R_m$ increased from 28.3% to 55.8%, which was the largest increase. The increase in ballast size had little effect on the $m_{total}$, but it considerably increased the $m_{residue}$ in the horizontal pipeline, which resulted in a significant increase in $R_m$. Therefore, an increase in ballast size leads to the risk of pipeline stagnation and blockage. Based on the research, the most likely area of blockage is the horizontal pipe section. Increasing the slurry flow rate can reduce the risk of blockage in the pipeline system, and increasing the ballast volumetric concentration can improve the carrying performance of the ballast.

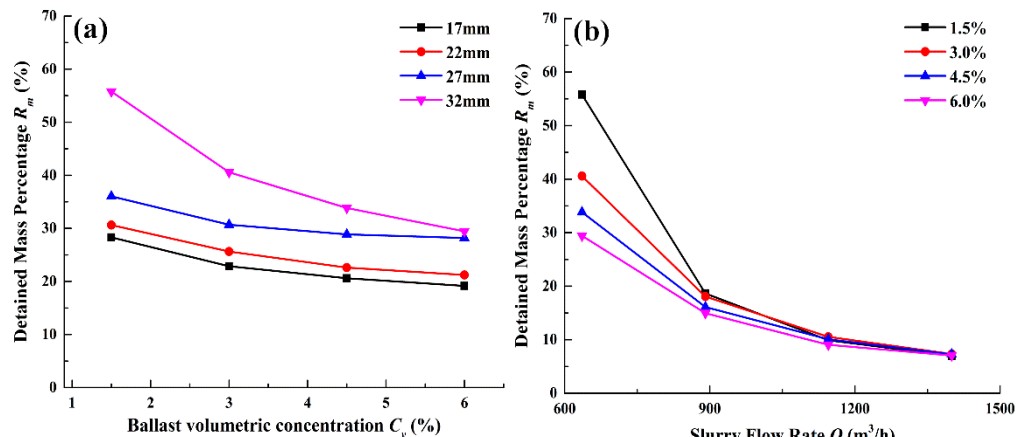

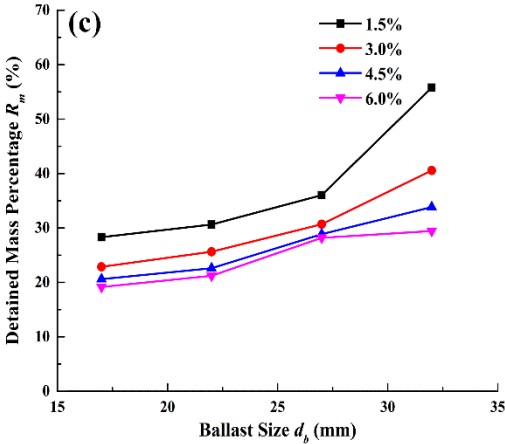

**Figure 12.** Diagram of the influence law diagram of ballast-detained mass percentage. The effect of (**a**) ballast volumetric concentration, (**b**) slurry flow rate, and (**c**) ballast size.

### 4.3. Influence Law of Ballast Distribution State in Each Pipe Section

We found that the ballast at $I_2$ of the inclined pipe is mainly located in the middle and upper part, the ballast at $H_2$ of the horizontal pipe is located in the bottom part, the ballast at $V_1$ of the vertical pipe is located on the flushing side, and the ballast at $V_2$ of the vertical pipe is in the center of the pipeline while using the statistics of the ballast distribution state at positions $I_2$, $H_2$, $V_1$, and $V_2$ under different working conditions. When the $d_b$ was 17 mm and the $C_v$ was 3%, the ballast distribution state under different slurry flow rates was as shown in Figure 13a. When the $Q$ exceeded 1145 $m^3/h$, the ballast at the $I_2$ of the inclined pipeline changed from bottom movement to suspension motion. However, it did not change the ballast motion state of the horizontal pipe at $H_2$, but the ballast distribution area reduced from regions 1–3 and 14–16 to regions 1–2 and 15–16. The ballast diffusion area at the $V_2$ of the vertical pipeline decreased from 60% to 24%. When the $d_b$ was 17 mm and the $Q$ was 891 $m^3/h$, the ballast distribution state under different $C_v$ was as shown in Figure 13b. When it increased from 1.5% to 6%, the ballast motion states in each pipeline were basically unchanged, but the ballast diffusion area in each pipeline increased. The ballast diffusion area increased along the left and right pipe sides at $H_2$ of the horizontal pipeline, and the accumulation area expanded from regions 1 and 16 to the regions 1–3 and 14–16. The ballast diffusion area extended along the slurry flushing side at the $V_1$ of the vertical pipe. The ballast diffusion degree at $I_2$ and $V_2$ increased obviously, the area proportion at $I_2$ increased from 36% to 59%, and the area proportion at $V_2$ increased from 17% to 60%. When the $Q$ was 891 $m^3/h$ and the $C_v$ was 1.5%, and the ballast distribution state under different ballast sizes was as shown in Figure 13c. When the ballast size was larger than 22 mm, the ballast at $I_2$ changed from suspension motion to bottom movement, and the ballast was mainly distributed in the bottom 1 and 16 areas. At the horizontal section $H_2$ and vertical sections $V_1$ and $V_2$, the diffusion degree and ballast motion state were basically unchanged.

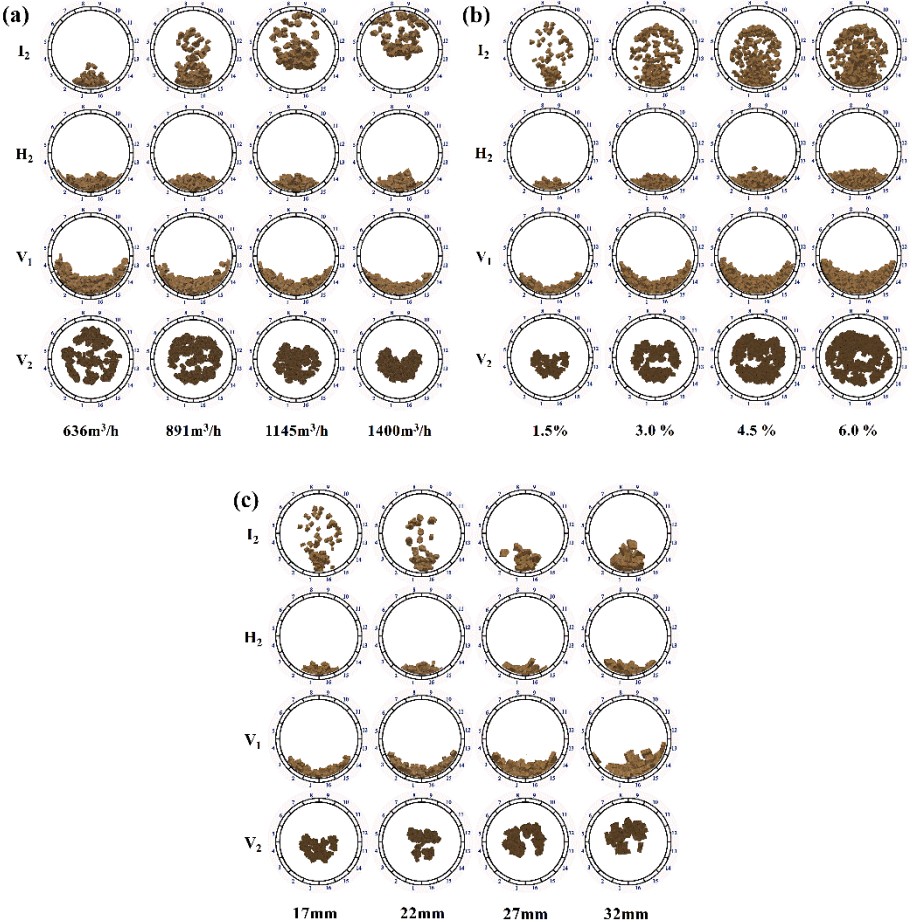

**Figure 13.** The ballast distribution state in each pipeline section of the slurry system. The effect of (**a**) ballast volumetric concentration, (**b**) slurry flow rate, and (**c**) ballast size.

## 5. Engineering Verification

In the above study, we analyzed the influence of slurry flow rate, ballast size, and ballast volumetric concentration on the mass flow rate, detained mass percentage, and ballast distribution state. It was not feasible to verify the influence laws of ballast mass flow rate and detained mass percentage in the numerical model at the construction site due to the construction risks and difficulties in data monitoring and information acquisition. However, we found that after a certain period of time, each pipeline section experiences different degrees of wear. During ballast transportation, a sliding friction sound is heard at the bottom of the pipe in the slurry system. The ballast size in the slurry system is greater than 15 mm and the slurry flow rate is about 3.5 m/s; compared with the research in literature [7,27], we judged that the ballast mainly moves at the bottom of the pipe. This indicates that the interaction between the ballast and the pipe inner wall during the motion process mainly causes the wear on the pipe. The pipe wear image was extracted on-site and the ballast distribution state from the simulation under the same working condition was acquired, as shown in Figure 14.

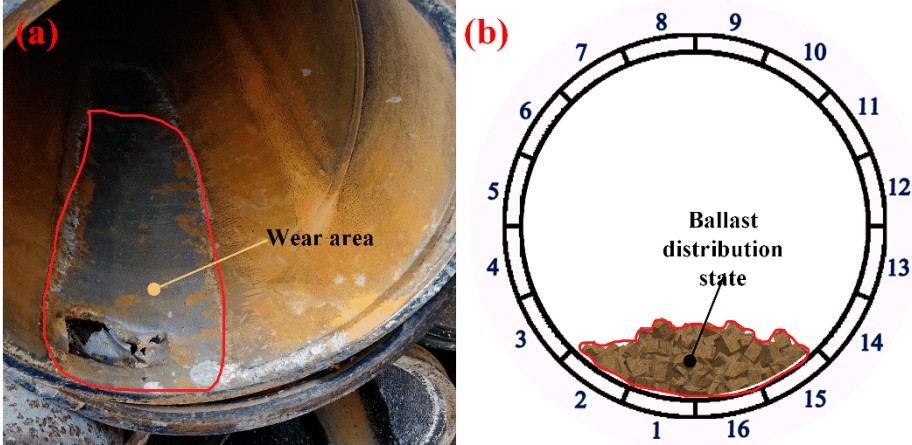

**Figure 14.** (**a**) Pipe wear image on-site; and, (**b**) the ballast distribution map of the pipe cross-section.

According to the field investigation above, Archard's wear theory could explain the pipe wear [28]. The wear model is:

$$M_v = KHF_n L_t \tag{9}$$

where $M_v$ is the volume of the steel pipe removed, $K$ is a dimensionless constant, $H$ is the hardness of the steel pipe, $F_n$ is the normal contact force from the ballast, and $L_t$ is the tangential displacement distance of the ballast.

Based on this theory, we concluded that the pipe wear degree is directly linked to the amount of the ballast, which infers that the pipe wear degree is higher in places with more ballast and lower in places with less ballast. The pipe wear degree can be obtained by measuring the thickness of the pipeline in various regions. The Model Smart AS860 ultrasonic wall thickness tester (SMART SENSOR, Dongguan City, China), maintaining an accuracy of 0.01 mm, was used to measure the wall thickness of 16 regions on the pipeline cross-section. Each region was measured three times and the average value was taken as the wall thickness of the pipeline in this region. The original wall thickness of the measured pipe was 12 mm. Through the areas where the wall thickness varied, the ballast distribution regions and wear degree were obtained. The three positions of each pipeline section were measured on-site and are marked in Figure 15b–d.

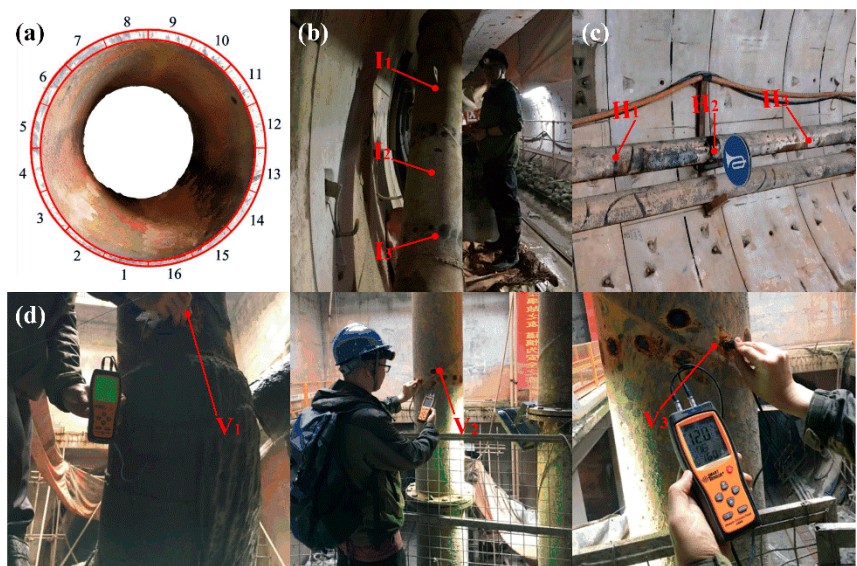

**Figure 15.** Pipeline measurement position on-site: (**a**) measuring area, (**b**) inclined pipeline, (**c**) horizontal pipeline, and (**d**) vertical pipeline.

### 5.1. Ballast Distribution Law in Inclined Pipeline

The inclined pipeline was used until the end of the project, and the wall thickness varying curves at $I_1$, $I_2$, and $I_3$ were measured, as shown in Figure 16. The wall thickness at $I_1$ in regions 1–3 and regions 14–16 was 10.80–11.80 mm, and the maximum wear rate was only 10.0%. The wear areas of $I_2$ and $I_3$ were basically the same as that of $I_1$, but the wear degree was significantly higher than that of $I_1$. The wall thickness of the wear area was 5.50–10.50 mm, and the maximum wear rate was 54.2%. The field condition corresponds to the simulation condition where the $Q$ is 891 m$^3$/h, the $C_v$ is 3%, and the $d_b$ is 17–32 mm. By extracting the ballast distribution regions at $I_1$, $I_2$, and $I_3$, we concluded that the ballast at $I_1$ is highly likely in a suspended motion state. The ballast contacts the pipe inner wall when the $d_b$ is larger than 22 mm, so the abrasion degree at $I_1$ is low. As the ballast movement distance increases, the ballast at $I_2$ and $I_3$ gradually changes from suspended motion to bottom movement. The ballast was mainly distributed in regions 1–3 and 14–16, and the distribution areas of the two positions were similar. The probability of the ballast contacting the inner wall is higher, so the wear degree at $I_2$ and $I_3$ increases. Therefore, we concluded that the ballast distribution regions and the pipeline wall thickness curves correspond to each other.

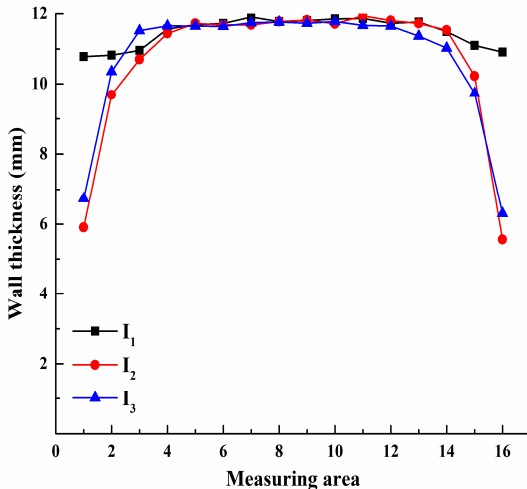

**Figure 16.** The wall thickness distribution curve of the inclined pipe section.

### 5.2. Ballast Distribution Law in Horizontal Pipeline

The wear of the horizontal pipeline is more severe than in other pipeline sections, and the pipe replacement frequency is high. The wall thickness curves at $H_1$, $H_2$, and $H_3$ were measured on-site, as shown in Figure 17. The laws of wall thickness at $H_1$, $H_2$, and $H_3$ are similar, and the wall thickness changing area was mainly located in regions 1–3 and 14–16. The wall thickness in regions 1–3 was 3.20–9.80 mm and the maximum wear rate was 73.3%. The wall thickness in regions 14–16 was 3.77–10.20 mm, and the maximum wear rate was 68.6%. The wall thickness in regions 4–13 was 11.10–11.90 mm, which is close to the original pipeline wall thickness, which indicates no wear. When compared with the inclined pipeline, as the variation in wall thickness of the horizontal pipe section increased, and the maximum wear rate increased by 19.1%. We determined that the ballast distribution states of $H_1$, $H_2$, and $H_3$ were similar by extracting the ballast distribution graph of the horizontal pipeline under simulation conditions. The ballast mainly accumulated in regions 1–4 and 13–16, and driven by bottom movement, which had a high probability of contact with the pipe inner wall. Therefore, the horizontal pipeline had a higher degree of wear. The ballast distribution under the simulation conditions was basically consistent with the actual measurement results.

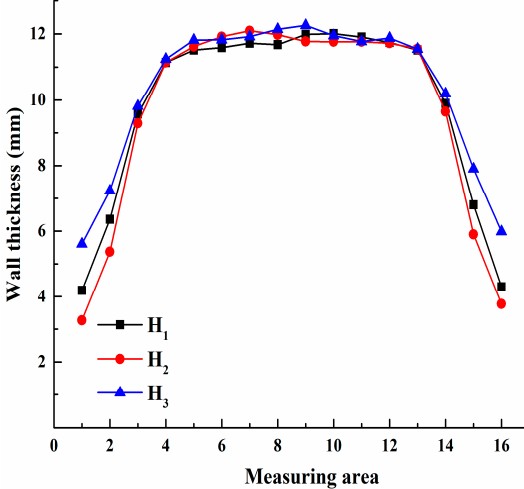

**Figure 17.** The wall thickness distribution curve of the horizontal pipe section.

### 5.3. Ballast Distribution Law in Vertical Pipeline

The distribution rule of the pipe wall thickness at $V_1$, $V_2$, and $V_3$ was obtained by measuring the vertical pipeline wall thickness, as shown in Figure 18. The wall thickness at $V_1$ mainly varied in regions 1–6 and 11–16, at 4.02–5.03 mm in regions 1–3 and 14–16, 6.65–10.90 mm in regions 4–6 and 11–13, and 10.30–11.30 mm in regions 7–10, with a maximum wear rate of 66.5%. The ballast at $V_1$ was mainly distributed in regions 1–6 and 11–16, which is the flushing side, whereas the regions 7–10 that deviated from the flushing side were less likely to contact the ballast. The wall thickness at $V_2$ and $V_3$ was evenly distributed, the measured wall thickness was 11.50 mm, and the maximum wear rate was only 4.2%, indicating that a low probability of contact between ballast and the pipeline inner wall at $V_2$ and $V_3$, with basically no wear. We concluded that the ballast at $V_1$ mainly concentrated in regions 1–5 and 12–16, which is smaller than the measured area, according to the ballast distribution maps of $V_1$, $V_2$, and $V_3$ under the corresponding simulation conditions. The number of ballasts in regions 1–3 and 14–16 was higher than in regions 4–5 and 12–13, whereas in regions 6–11, no ballast was distributed. In this case, the wall thickness of the pipeline increased gradually from regions 1 to 6, but gradually decreased from regions 7 to 12. The ballast at $V_2$ and $V_3$ gathered toward the central area of the vertical pipeline, and a certain distance exists between it and the pipeline inner wall, so the wall basically has no wear. The distribution law of ballast in the vertical pipeline section was also consistent with the actual situation.

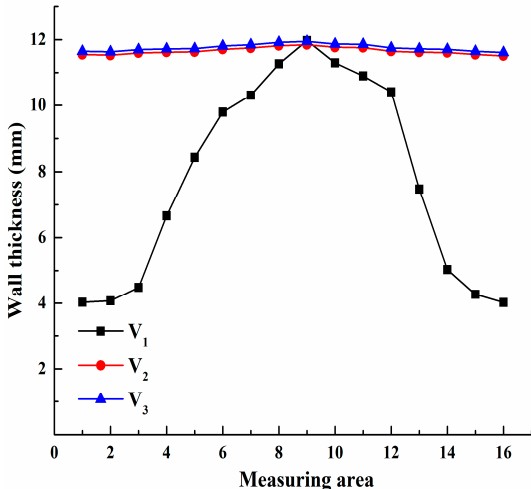

**Figure 18.** The wall thickness distribution curve of the vertical pipe section.

## 6. Conclusions

The ballast carrying capacity and movement state in the slurry system was studied based on a two-way coupling of the discrete phase model, and the conclusions are as follows:

(1) A hydraulic lifting numerical model of a four-meter-long vertical pipeline was built with the spherical ballast group with particle size between 10 and 40 mm as the research object. The accuracy of the numerical model was verified by comparing the distribution state and suspension speed of ballast, and the difference between experimental and simulation data was within 8%.

(2) Increases in the slurry flow rate and ballast volumetric concentration improve the ballast mass flow rate $K_m$ in each pipeline and reduce the detained mass percentage $R_m$ in the horizontal pipeline, to effectively improve the carrying capacity of the pipeline. With increasing slurry flow rate, the maximum increase in $K_m$ was 120%, and 322% when the ballast volumetric concentration increased. The effect of the ballast volumetric concentration on the mass flow rate is stronger than that of the slurry flow rate. With increasing slurry flow rate, the maximum decrease of $R_m$ was 87.5%, whereas the maximum decrease of $R_m$ was 47.3% with the increase in the ballast volumetric concentration. The slurry flow rate has a stronger effect on the $R_m$ than ballast volumetric concentration. Increases in ballast size have no effect on the $K_m$ of each pipeline section, but they can significantly increase the $R_m$ of the horizontal pipeline section with a maximum increase of 97.2%. Therefore, increasing the ballast size is likely to cause a risk of the ballast retention and blockage in the horizontal pipeline.

(3) The slurry flow rate and the ballast size are the main factors affecting the ballast motion state. Increasing the slurry flow rate changes the ballast movement state to suspension in inclined pipelines, decreases the ballast accumulation area in horizontal pipelines, and aggregates the ballast in the vertical pipelines toward the central area. Increasing the ballast size changes the ballast motion state to bottom movement in the inclined pipeline, and it increases the ballast accumulation area in the horizontal pipeline, which has little effect on the diffusion degree and ballast movement state in the vertical pipeline. The increase in the ballast volumetric concentration has little effect on the ballast motion state in each pipeline. It can significantly increase the ballast diffusion area in the inclined and vertical pipelines, and increase the ballast accumulation area in the horizontal pipeline.

(4) We found that the variation in the wall thickness in various regions of the inclined section and the horizontal section was mainly concentrated in the bottom of the pipeline by measuring the wall thickness of each pipeline in the field. The maximum wear rate of the inclined pipeline was 54.2%, whereas that of the horizontal pipeline was 68.6%. The horizontal pipeline wall thickness was basically unchanged along the length direction. The areas where wall thickness varied at $V_1$ of the vertical pipeline were mainly located on the flushing side, and the maximum wear rate was 66.5%. The inner wall of the vertical pipe at $V_2$ and $V_3$ did not wear, and the maximum wear rate was only 4.2%. The wall thickness variation curve and the ballast distribution area in the simulation condition were mutually proved.

**Author Contributions:** Y.X. and G.Z. is responsible for the frame design of the paper and guide the writing of the paper. Y.W. performed experiments and numerical studies on analytical objects and write the paper. X.X. is responsible for the model setup and helps analyze the data. H.X. and P.C. provide on-site construction data and assists in the survey.

**Funding:** The research is supported by the National High Technology Research and Development Program of China (Grant No 2012AA041803), Science and Technology Major Project of Hunan Province, China (Grant No 2014FJ1002), and the Fundamental Research Funds for the Central Universities of Central South University (Grant No 2017zzts089). The supports are gratefully acknowledged by the authors.

**Conflicts of Interest:** The authors declare no conflict of interest.

## Abbreviations

| Symbol | Description |
|---|---|
| $\alpha_f$ | slurry volume fraction |
| $\rho_f$ | slurry density |
| $u_f$ | slurry flow velocity |
| $\tau_f$ | slurry shear stress |
| $p$ | slurry pressure |
| $g$ | acceleration of gravity |
| $F_j^f$ | ballast and slurry interaction force |
| $k$ | slurry viscosity coefficient |
| $n$ | power law exponent |
| $\varphi_b$ | ballast shape coefficient |
| $A_S$ | sphere surface area |
| $A_b$ | ballast surface area |
| L | length |
| W | width |
| H | height |
| $d_e$ | ballast equivalent diameter |
| $V_b$ | ballast volume |
| $A_S$ | sphere surface area |
| $A_b$ | ballast surface area |
| $\varphi_b$ | ballast shape coefficient |
| $m_j$ | ballast mass |
| $u_j$ | ballast translational velocity |
| $\omega_j$ | ballast rotational velocity |
| $I_j$ | ballast inertial mass |
| $F_{c,i}^j, F_{c,j}^w$ | contact force |
| $F_{d,i}^j, F_{d,j}^w$ | viscoelastic force |
| $T_i^j, T_w^j$ | torque |
| $T_{DT}^j$ | ballast motion moment |
| $F_D$ | drag force |
| $F_P$ | pressure gradient force |
| $F_{vm}$ | virtual mass force |
| $F_M$ | Magnus rotating lifting force |
| $C_j$ | slurry–ballast exchange coefficient |
| $C_{vm}$ | virtual mass factor |
| $C_{RL}$ | rotational lift coefficient |
| $\Omega$ | relative angular velocity |
| $A_p$ | projected ballast surface area |
| $D_p$ | pipeline diameter |
| $L_1$ | inclined pipeline length |
| $L_2$ | horizontal pipeline length |
| $L_3$ | vertical pipeline length |
| $\alpha$ | inclined angle |
| CFD | computational fluid dynamics |
| DEM | discrete element method |
| $d_b$ | ballast size |
| $\rho_b$ | ballast density |
| $Q$ | slurry flow rate |
| $C_v$ | ballast volumetric concentration |
| $K_m$ | ballast mass flow rate |

| $R_m$ | detained mass percentage |
| $m_{residue}$ | ballast remaining quality |
| $m_{total}$ | ballast total quality |

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
