# Peer review of "Ballast Flow Characteristics of Discharging Pipeline in Shield Slurry System"

_applsci, doi:10.3390/app9245402_

Round 1

Reviewer 1 Report

In the present work, a rather interesting study of ballast flow is provided.

In the reviewer’s opinion, before getting accepted for publication the following issues need to be accurately accounted for:

Grammar must be improved quite a lot. The abstract is excessively long and must therefore be reduced and better focused. 1 does not seem clear and representative enough for people not already acquainted with the problem. It is suggested to provide a better one, especially in terms of the real flowing process. Line 81: what is the meaning of particle size 0 mm? There should be a minimum, non-zero value. At the end of the Introduction, a brief paragraph listing the contents of the whole paper should be introduced. All the acronyms must be defined, even of common use. Notation in Eq. (2) should be checked: is the shear stress a scalar or vectorial field? In the first case, the dot to denote the divergence (we guess) is not needed; in the second case, bold symbol must be used. (3): what is the difference between this shear stress and that appearing in Eq. (2)? Lines 135-136: values of the coefficients should be better placed in a Table, for a general reference. Line 143: is that value 80% or 90%? There seems to be a discrepancy between the text and the plot. Line 144: why a discrepancy or fluctuation in the value of the density? Please, elaborate on it. Line 149: for self-completeness, please add a definition of the shape coefficient. 4: are the numerical models of the ballast samples 2D or 3D? Line 191: is here the size or length of the pipeline, of interest? Along the same line, how are size effects taken into account? Lines 201-202: according to what stated here, it means that elements must be big (bigger than the biggest particle). How can this guarantee accuracy, as usually small elements are required? Line 205: which explicit time integration procedure was adopted? How was conditional stability guaranteed? Lines 207-209: moving from 50 s analysis time for each group to the mentioned 45 days in total is not clear at all. Explain better. Section 3: before moving to numerical model validation, one is supposed to read something about details of the model itself, which are not provided. How are FEM and DEM analyses coupled? If you think it becomes too heavy within the text, an appendix can be added with everything necessary to understand the method. Line 226: here ballast density changes with respect to the values reported at line 144. Why? Line 256: check the usage of sub- and super-script within the entire text. Lines 263-270: the entire paragraph should be re-phrased to explain this issue much better. Lines 285-286: how is it possible to judge the present statement, if only the results relevant to one value of the size are reported? 11: here and also in other ones, try to avoid the use of acronyms in the legends. Lines 389-390: what is the expected scattering in the thickness, out of the production process? Section 5 just provides a purely qualitative validation of the numerical results. This is to be explicitly stated, otherwise people may expect to find also a quantitative validation. A wear model should be also added to somehow explain the measured values of the thickness; this will allow to provide a link with the presented numerical results at the end of the conclusion, see Lines 499-508.

Reviewer 2 Report

The paper is clear and well written.

The topic appropriate for the Journal.

The papper was correctly edited and graphically developed at a goog level.

I do not see any shortcuts.

The paper deserves a positive assessment because it is current and interesting from  both a cognitive and practical point of view.

Author Response

Dear Reviewer,

Thank you very much for your approval and support of my paper.

Reviewer 3 Report

This paper definitely would benefit if more detailed information about the novelty of the proposed approach was included in the text based on the literature available so far. As the conclusions are quite physically clear it may be assumed that the proposed approach is correct. 

From an editorial point of view, I would pay more attention to the location of the figures captions.

Author Response

Dear Reviewer,

Thank you very much for your reply and help. The deficiencies of the paper have been improved with your help. 

Round 2

Reviewer 1 Report

The authors have appropriately replied to all my comments.